# Deep Adversarial Forward Model

## Abstract

Learning world dynamics has recently been investigated as a way to make reinforcement learning (RL) algorithms to be more sample efficient and interpretable. In this paper, we propose to capture an environment dynamics with a novel forward model that leverages recent works on adversarial learning and visual control. Such a model estimates future observations conditioned on the current ones and other input variables such as actions taken by an RL-agent. We focus on image generation which is a particularly challenging topic but our method can be adapted to other modalities. More precisely, our forward model is trained to produce realistic observations of the future while a discriminator model is trained to distinguish between real images and the model's prediction of the future. This approach overcomes the need to define an explicit loss function for the forward model which is currently used for solving such a class of problem. As a consequence, our learning protocol does not have to rely on an explicit distance such as Euclidean distance which tends to produce unsatisfactory predictions. To illustrate our method, empirical qualitative and quantitative results are presented on a real driving scenario, along with qualitative results on Atari game Frostbite.

## 1 Introduction

The capability to predict the consequences of actions is an inherent part of biological decision making Clark (2013). Though this capacity is not always required, it becomes useful for tasks that require planning and more generally forecasting ahead of the current situation. Such a capability has been shown to be particularly useful in the case of environments with complex dynamics (Bertsekas, 1995). Indeed, modeling the dynamics of the surrounding world has led to significant improvements in the decision process thus providing the ability to anticipate (Weber et al., 2017; Silver et al., 2017).

In the literature of decision and control, an environment forecasting model is commonly called a *Forward Model* Henaff et al. (2017); Pathak et al. (2017). Such a model estimates what the future observation will be with respect to the current one and possibly other conditioning variables. These models are trained in a supervised fashion, either online or offline. More recently, several papers have shown that forward modeling can be used to quantify the inherent curiosity of a learning decision policy. Curiosity is defined as the error of a forward model during the course of an episode in a considered environment. Such quantity has been recently used to maintain the explorative behave of a decision policy and has been shown to be more efficient than maintaining an entropy regulariser as an egocentric reward.

In this paper, we introduce an algorithm that learns forward models without relying on any kind of task-specific loss. Our method leverages recent improvements in adversarial training Goodfellow et al. (2014); Chen et al. (2016a). As far as our knowledge goes, it is the first attempt to use adversarial learning for such a forward modeling task. More precisely, we train a forward model in the form of a neural network, in an unsupervised fashion, through a discriminator network which has to distinguish between true and generated samples. By using this adversarial learning paradigm, the need to find the appropriate loss for each specific task is removed. Furthermore, this learning approach does not put any constraint on the modality of the observations the model receives and generates. The rest of the paper is organized as follows, after presenting the main elements of the state of the art, we describe the proposed methods and model. Then, we detail the experiments and the encouraging associated results that demonstrate the benefits of the proposed method.

## 2 RELATED WORK

In decision theory, forward models are commonly defined as auto-regressive architectures which can be trained using collected agent trajectories. Given a past state $s_t$ and current action $a_{t+1}$, the model predicts the next state $p(s_{t+1}|s_t, a_{t+1}) = f_\theta(s_t, a_{t+1})$ of the environment, with $f_t heta$ a function, for example a neural network, parameterized by $\theta$. As an alternative definition, other signals such as reward can also be taken into account. Several arguments support the learning of a forward model (Bertsekas, 1995) such as easier control policy definition as an optimization problem. More recently, several papers have already proposed deep learning for such a task (Leibfried et al., 2016; Oh et al., 2015). However, these works are currently using either handcrafted or pixel-wise representations to derive a computable and differentiable loss. More generally, several fields of research can dramatically benefit from this forward modeling.

**Exploration efficiency**: Curiosity-driven exploration is formulated as the error in a learning agent's ability to predict the consequence of its own actions (Pathak et al., 2017). In this context, this error is measured through a forward model of the considered environment. The main motivation of such a measurement used as an intrinsic reward is to cope with the sparsity of extrinsic reward. It has been recently studied in the domain of visual control in grid worlds and Atari 2600 video games Henaff et al. (2017). Such a metric is used to encourage exploration during the policy learning process. Moreover, the prediction error has also been used in the feature space of an auto-encoder as a measure of interest of a state to explore (Stadie et al., 2015).

**Imagination-based control**: Predicting the sequence of future outcomes from a given action has been recently investigated as an additional source of information for complex decision making. Indeed, learning *World Models* (Ha & Schmidhuber, 2018) has been proposed as a promising way to decrease the complexity of a policy model. The idea is to delegate a part of the overall control learning task to a dynamic model instead of learning the environment representation and the optimal policy in a monolithic model trained in an end-to-end fashion. Thus, it becomes possible to benefit from environment observations that are not necessarily correlated with the task and the controller. Moreover, such a technique has been investigated as a way to perform transfer learning through tasks given an environment dynamic (Kansky et al., 2017). The *predictron* (Silver et al., 2017) is a recent example of such an approach. This policy network learns environment dynamics as part of its component to enrich its decision support. Imagination-based control has provided encouraging initial results (Weber et al., 2017). However, the end-to-end nature of the learning setup proposed in these works makes the interpretability of the model difficult.

**Interpretability**: By developing parts of a decision model that can directly be observed - for example by looking at generated images of the future - it becomes possible to improve the interpretability of the model (Maes et al., 2012). Indeed, forward modeling makes it possible to observe the decision support of the policy compared to implicit representations that only the model can use.

For these reasons, we posit that forward modeling is an important field of research in the domain of control. In this paper, we propose to leverage the recent advances in imagination-based control and conditional generative adversarial models along with inverse modeling to develop a novel and generic forward model that is able to cope with synthetic and realistic visual control scenarios.

## 3 DEEP ADVERSARIAL FORWARD MODEL

### 3.1 PROBLEM FORMULATION

We consider a transition function $\mathbb{T}$ that maps the current observation $s_t$ to the the next observation $s_{t+1}$, i.e., $s_{t+1} = \mathbb{T}(s_t, a_t)$ where $a_t$ the action taken by an agent. A forward model $f_\theta$ tries to approximate the transition function $\mathbb{T}$ and thus give an estimate $\widehat{s}_{t+1}$, such that $\widehat{s}_{t+1} = f(s_t, a_t)$ is close to $s_{t+1}$. The task of learning the forward model consists of learning the parameter $\theta$ that will minimize the error between $s_{t+1}$ and $\widehat{s}_{t+1}$.

### 3.2 MODEL ARCHITECTURE

The forward model is made of two neural networks. Figure 1 shows the architecture of the forward model. First, the observation $s_t$ is encoded through a pretrained ResNet34 He et al. (2016) that

aims at compressing the information in the raw observation. This abstract representation $\Phi(s)$ of the raw observations follows the approach proposed by Pathak et al. Pathak et al. (2017). Then, this representation is concatenated with the action $a_t$ taken at this time step to obtain $c_t$. This is necessary because, in many situations, the current observation is not a sufficient piece of information to predict the next observation. For example $a_t$ can represent the position of the wheel and velocity in the case of a forward model for driving. Finally, this conditioning $c_t$ is fed to a generator network $G$ to obtain an estimate of the observation at the next timestep:

$$\widehat{s}_{t+1} = f_\theta(s_t, a_t) = G(\Phi(s_t), a_t) = G(c_t). \tag{1}$$

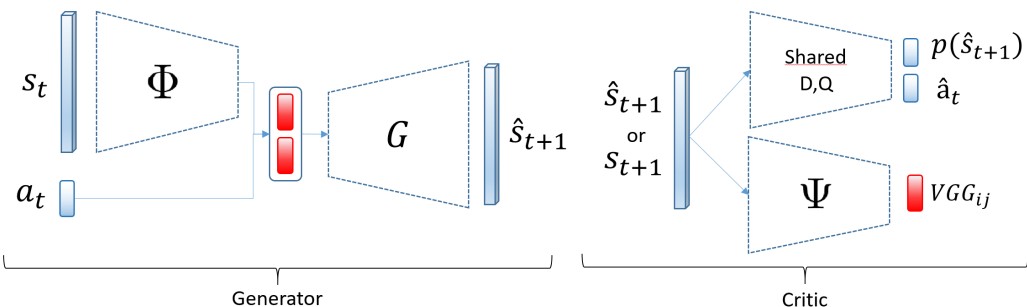

Figure 1: General figure showing the forward model architectures and the networks used for training.

## 3.3 Training and loss Functions

Our forward model is trained using a linear combination of three losses applied on the general architectures shown on figure 1. The main loss is an *adversarial loss* encouraging the forward model to produce realistic forward estimates $\widehat{s}_{t+1}$. This loss is complemented with a *mutual information loss* for encouraging the forward model to make use of the conditioner. Finally, a *feature space loss* on a pretrained convolutional network encourages visual content to be similar in $\widehat{s}_{t+1}$ and $s_{t+1}$. To the best of our knowledge, it is the first composition of such losses schema in this context of forward modeling. We detail these losses below.

**Adversarial Loss:** First, we want to generate forward samples that are realistic enough with regard to the current observation. For this, we follow the standard formulation of Generative Adversarial Learning Goodfellow et al. (2014). In the adversarial setting, the role of discriminator $D$ is to provide the loss function for training the generator $G$ which correspond to the actual forward model we want to learn. To do this, the discriminator network $D$ learns to distinguish the real next observation $s_{t+1}$ from its estimate $\widehat{s}_{t+1}$ produced by $G$. The output of $D$ is used as a probability of its input being real versus generated by $G$. At the same time, $G$ is trained to fool $D$, i.e., it is trained to generate samples that are hard for $D$ to distinguish from the real ones. This known paradigm leads to the following two-player min-max game with value function $V(G, D)$:

$$\min_G \max_D V(G, D) = \mathbb{E}_{x \backsim p_{\text{data}}}[\log(D(x))] + \mathbb{E}_{z \backsim p_{\text{noise}}}[\log(1 - D(G(z)))]. \tag{2}$$

**Mutual-Information Loss:** Secondly, we need to encourage the generator to use all the provided auxiliary information $c_t$ to generate samples that take into account the current conditioner. For this, we introduce an additional loss $\mathcal{L}_c$ inspired from the InfoGAN setup Chen et al. (2016b). An additional network $Q$ - which in practice shares most of its layers with $D$ - outputs an estimation $\widehat{c}_t$ of $c_t$. The associated loss $\mathcal{L}_c$ is defined as the cross-entropy between $\widehat{c}_t$ and $c_t$ in the case of discrete distributions and the mean squared error between them in the case of continuous ones. The addition of the mutual information loss turns the GAN objective into $\min_{G,Q} \max_D V_{\text{InfoGAN}} = V(D, G) - \lambda \mathcal{L}_c$.

In reinforcement learning, such a loss is used to train an *inverse model*. The inverse model model tries to guess which actions cause the transition from $s_t$ to $s_{t+1}$. Here, we just use this loss as auxiliary information to constrain the generator to make usage of the conditional inputs $c_t$.

**Content Loss:** Finally, in order to ensure that the generator outputs visually similar content than the one in $s_{t+1}$, this third loss computes a similarity information based on the aspect of the generated visual states. A naive approach would be to add a weighted pixel-wise loss function such as Mean Squared Error (MSE). In practice, these kinds of losses are very prone to generate blurry samples as they encourage the textures to be overly smooth, thus leading to bad image quality. Recently, Ledig et al. (Ledig et al., 2016) showed that combining both adversarial and content losses greatly help the generator to output realistic samples by relying on an abstract representation of the image. Following this, we include a content-loss $\mathcal{L}_\Psi$ that computes the Euclidean distance between the output representation generated by a network $\Psi$ for $s_{t+1}$ and $G(\Phi(s_t), a_t)$. In our specific setup, we adopt the method (Ledig et al., 2016) to use $VGG_{ij}$ for $\Psi$, where $VGG_{ij}$ means the feature map obtained after the j-th convolution (after activation) and before the i-th max-pooling layer in the VGG19 network (Simonyan & Zisserman, 2014). Thus we define $\mathcal{L}_\Psi$ as

$$\mathcal{L}_\Psi = \frac{1}{C_{i,j}W_{i,j}H_{i,j}} \sum_{c=1}^{C_{i,j}} \sum_{x=1}^{W_{i,j}} \sum_{y=1}^{H_{i,j}} (\Psi(\widehat{s}_{t+1})_{cxy} - \Psi(s_{t+1})_{cxy})^2. \tag{3}$$

Our final loss $\mathcal{L}$ is defined as a linear combination of all the aforementioned individual losses:

$$\mathcal{L} = \alpha\mathcal{L}_{adv} + \beta\mathcal{L}_c + \gamma\mathcal{L}_\Psi.$$

## 3.4 CONDITIONING THE FORWARD MODEL

In order to generate context-dependent samples, we follow the general idea of InfoGAN (Chen et al., 2016a), illustrated in Figure 2 where the generator is provided extra latent-codes $c$ defined as the concatenation of latent-variables $\{c_1, \ldots, c_L\}$. In the case of forward-modeling, this proposed approach of generative modeling makes sense as it allows to properly condition the next observation generation in a metric-free loss manner.

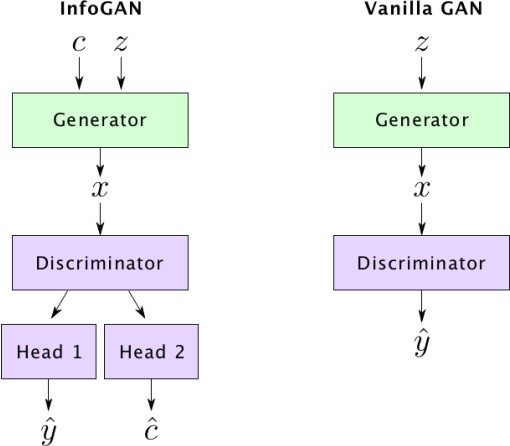

Figure 2: InfoGAN vs Vanilla GAN

To the best of our knowledge, this is the first proposal of using such a framework of conditional adversarial training for forward-modeling. In this framework, an information-theoretic regularization enforces high mutual information between latent-codes $c$ and the generator distribution $G(z, c)$ in an unsupervised way. The mutual information $I(X, Y)$ between two random variables $X$ and $Y$ can be expressed as the difference of two entropy terms:

$$I(X, Y) = H(X) - H(X|Y) = H(Y) - H(Y|X).$$

The InfoGAN objective is:

$$\min_{G,Q} \max_{D} V_{\text{InfoGAN}}(D, G, Q) = V(D, G) - \lambda\, I(c; G(z, c)) \tag{4}$$

$I(c; G(z, c))$ takes the form $I(c; G(z, c)) = H(c) - H(c|G(z, c))$ and we aim at maximizing it. Because the posterior $P(c|G(z, c))$ is hard to estimate, the authors of InfoGAN suggest the use of a variational approximation, detailed in the original paper:

$$L = \mathbb{E}_{c \sim P(c), x \sim G(z,c)}[\log Q(c|x)] + H(c) \tag{5}$$

$$I(c; G(z, c)) \geq L \tag{6}$$

where $Q$ also takes the form of a neural network. In practice, $Q$ uses the same parameters as $D$, except for the last layer. $L$ is a lower bound on $I(c; G(z, c))$, the variational approximation consists of maximizing $L$ as a proxy to $I(c; G(z, c))$. The bound is tight as $\mathbf{Q}$ approaches the true posterior $P(c|G(z, c))$. The infoGAN training criterion becomes:

$$\min_{G,Q} \max_{D} V_{\text{InfoGAN}}(D, G, Q) = V(D, G) - \lambda\, L(G, Q) \tag{7}$$

## 4 EXPERIMENTS

To measure the effectiveness of our method, we use two approaches. First, following the evaluation framework suggested by Ledig et al. (Ledig et al., 2016), we performed quantitative evaluation of the importance of each sub-loss using two different evaluation criteria: the raw pixel-space distance and the distance in the latent $VGG_{54}$ feature space. The latter space is used to emphasize the importance of getting the more important parts of the picture right. Then, we present some qualitative results to illustrate the relevance of the generated samples with respect to the ground-truth images.

### 4.1 DATASET

We choose to evaluate our model on synthetic and realistic visual control settings. First, we selected a set of Atari2600 games to illustrate the capability of the proposed model and learning protocol to capture the environment of dynamic and transcribe it back into visual observations. In this case, the image is centrally cropped with a format of $1 \times 64 \times 64$ as the RGB channels are grayscaled. The action set is a discrete variable of 16 dimensions. As a second experiment, the Udacity dataset is a set of sequential $1 \times 128 \times 128$ images recorded by a dash-camera in front of a moving car. This dataset is originally used in the context of supervised autonomous driving learning tasks. In addition to the images, the dataset provides for each image the current speed, torque, and steering angle of the car. The goal here is to generate the next camera observation with respect to the current one and the speed and steering angle.

### 4.2 NETWORK & TRAINING DETAILS

In this section, we present the parameters of the model and the details of the learning algorithm. All experiments have been run on NVIDIA $4 \times V100$ GPUs using CUDA9.1 and CuDNN7.

**The generator network** receives as input a vector composed of the concatenation of $\Phi(s_t)$ and $c_t$ and outputs $3 \times 128 \times 128$ images. First, a linear layer converts this input to a dense vector reshaped as an initial $64 \times 4 \times 4$ image. Then, we use a sequence of 5 residual blocks with respectively $64, 128, 256, 128, 64$ channels. Upscaling is done as part of each residual block through a PixelShuffling layer (Shi et al., 2016) with upscaling factor of 2. We use batch normalization after each residual block and Leaky-ReLU non-linearity for each convolution layer.

**The discriminator network** follows the topology of DCGAN (Radford et al., 2015). It is made of 5 convolution layers with $32, 64, 128, 256, 512$ channels respectively. The kernel size is 4 with a stride of 2. We set the batch size to 128 samples and apply linear decaying dropout starting at $0.5$ for the first 250 epochs.

### 4.3 QUANTITATIVE RESULTS

In order to measure the contribution of our proposed set of losses, we compare our model in two different settings on the Udacity dataset. First, we trained the generator without the discriminator part, only relying on a pixelwise distance loss $\mathcal{L}_2$ between $\widehat{s}_{t+1}$ and $s_{t+1}$. This setting corresponds

to the model used in (Oh et al., 2015) for their forward model. Secondly, we use the architecture proposed in this paper, following the adversarial training loss $\mathcal{L}_{adv}$. Finally, we define $e_{pw}$ and $e_{VGG_{54}}$ respectively as the Euclidean distance in the raw pixel space and the Euclidean distance on the output of $VGG_{54}$ between the real and the generated images, comparing the results for all models.

| | Type of training loss | | | | |
| | $\mathcal{L}_2$ | $\mathcal{L}_{adv}$ | $\mathcal{L}_{VGG_{54}}$ | $\mathcal{L}_{adv} + \mathcal{L}_c$ | $\mathcal{L}_{adv} + \mathcal{L}_c + \mathcal{L}_{VGG_{54}}$ |
|---|---|---|---|---|---|
| $e_{pw}$ | 285.40 | 308.03 | 414.14 | 297.67 | **279.40** |
| $e_{VGG_{54}}$ | 20.70 | 21.72 | 18.85 | 21.73 | **19.14** |

Table 1: Experiment results in different training setups

Table 1 shows the values obtained for $e_{pw}$ and $e_{VGG_{54}}$ for the different configurations of training losses. From the results, we can see that the adversarial loss does not get all the information required to properly model the dynamics of the environment. Also, when combining the adversarial loss with the mutual-information loss $\mathcal{L}_c$, pixel recovering improves, but the content metric $e_{VGG_{54}}$ does not improve. Finally, combining all those three losses provides a valuable gain, both in terms of pixel values and content recovering.

In the next section, we present qualitative results on Frostbite in the Atari simulator. We choose Frostbite as it includes cases where an agent's actions have a direct impact on the environment, but also, cases where the agent has no control over the environment (like snow melting).

### 4.4 QUALITATIVE RESULTS

In this section part, we follow the customary GAN literature to include some qualitative results for illustration.

**Atari :**We trained an agent using Actor-Critic network Mnih et al. (2016) in order to play the game for one day. In parallel, we used the LSTM output and the action sampled from the Actor-Critic as a conditioner for our generator. As expected, our approach seems to produce sharp forward visual states. Furthermore, both the moving elements belonging to the environment dynamic and the game elements, which are action dependant, the hero representation, in this case, are reconstituted.

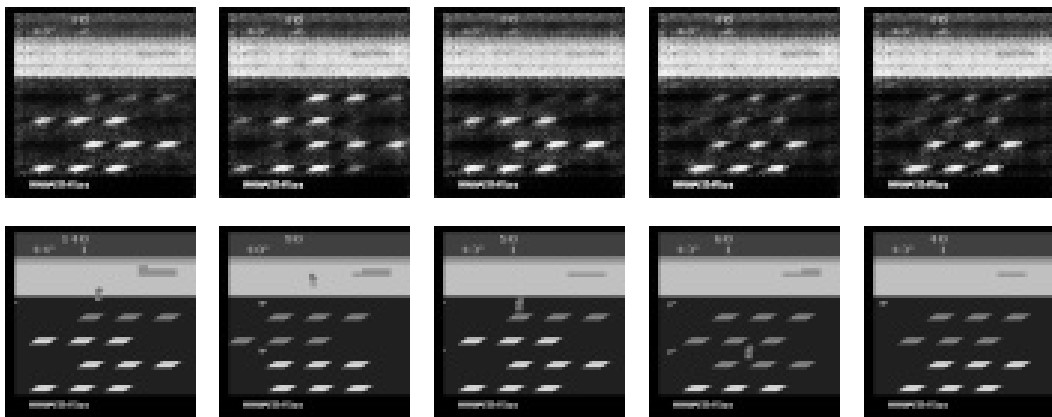

Figure 3: The first row represents the generated samples sequence. The second one is the real sequence.

**Udacity :** In this dataset, the images are downscaled to $128 \times 128 \times 3$ tensors and we use a ResNet-34 model as $\Phi(s_t)$ to encode the observation. Speed and steering angle are rescaled into range $[-1, 1]$. The proposed approach, the one on the top-most left column seems to produce the sharpest and clearest images. Indeed, the light changes, the road borders, textures and object silhouettes as the trees seem to be more in line with the original image.

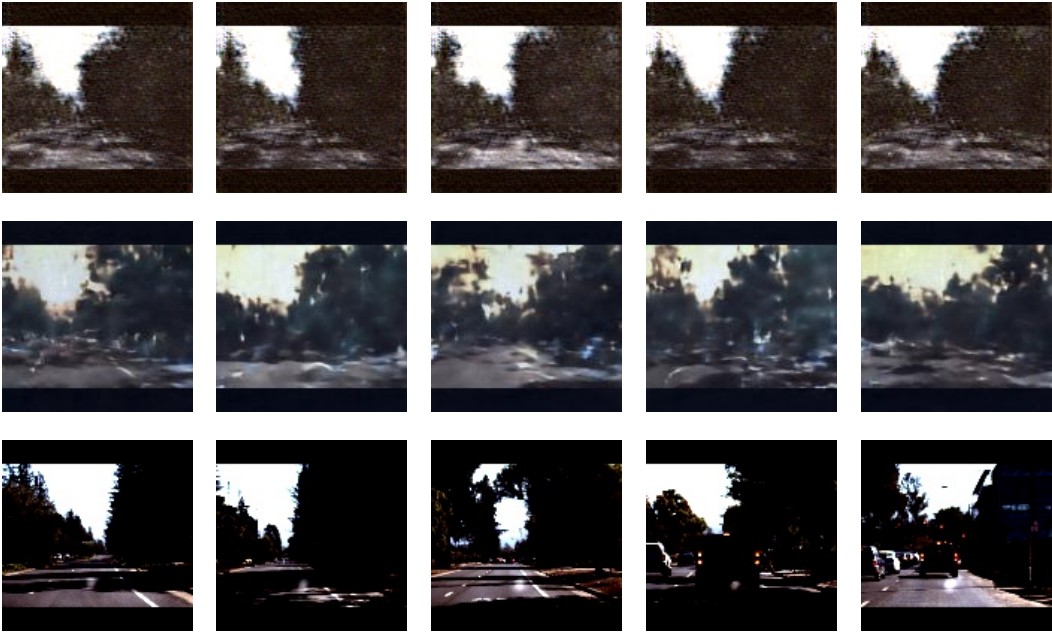

Figure 4: First row represents the generated samples with Adversarial, Mutual Information, and Content losses. Second one is samples generated by pixelwise loss. Finally, the last row presents the ground truth images

## 5 CONCLUSION

In this paper, we propose a novel method and a model that produces realistic samples of the observation of a given environment conditioned by the previous observation and action. This so-called forward-model is generic and imposes no specific restrictions over the modalities used as input. Moreover, in the case of complex structures, our method generates samples in the raw observation space without of feature engineering over the considered state space. The results provided by our approach allow for better interpretability in the case of a planning-based control over this forward model. An immediate extension of our setup could be to give more context to the generator by relying on sequential observations as decision support for the next frame generation. One another potential use of such a model is to guide the exploration in reinforcement learning.

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
