# OpenReview forum: "DEEP ADVERSARIAL FORWARD MODEL"
_ICLR.cc/2019/Conference_

### Official Review · AnonReviewer1 · 2018-10-30
**Interesting but not novel; could be better evaluated.**

**Rating:** 4
**Confidence:** 4

**Review:**

This paper describes an approach for training conditional future frame prediction models, where the conditioning is with respect to the current frame and additional inputs - specifically actions performed in a reinforcement learning (RL) setting.

The authors suggest that one can predict future frames from a vector comprised of an observation encoding and an action. To train the model, they suggest using a linear combination of three different losses: (1) an adversarial loss that encourages the generated sample to look similarly to training data, (2) an InfoGAN-inspired loss that is supposed to maximise mutual information between the conditioning (e.g. action) and the generated sample, and (3) a content loss, taken to be the mean-squared error of the prediction and ground-truth in the VGG feature space.

The major contribution of this work seems to be using these three losses in conjunction, while doing conditional frame prediction at the same time. While interesting, there exist very similar approaches that also use adversarial losses [1] as well as approaches using different means to reach the same goal [2, 3]. None of these are mentioned in the text, nor evaluated against. It is true that [1] is not action-conditional, but adding actions as conditioning could be a simple extension.

Experimental section consists of an ablation study, which evaluates importance of different components of the loss, and a qualitative study of model predictions. With no comparison to state of the art (e.g. [1, 3]), it is hard to gauge how valuable this particular approach is.
The qualitative evaluation starts with §4.4¶1 “we follow the customary GAN literature to include some qualitative results for illustration”, as if there was no other reason for including samples than to follow the custom. Since the paper is about action-conditional prediction, it would be interesting to see predictions conditioned on the same initial sequence but different actions, which are not present, however. Moreover, this work is developed in the context of RL applications, and since prior art [4] has shown that better predictive models do not necessarily lead to better RL results, it would be interesting to evaluate the proposed approach against baselines in an RL setting.

The paper is clearly written, but some claims in the text are not supported by any citations (e.g. §1¶2 “More recently, several papers have shown that forward modelling…” without a citation).  Some claims are misleading (e.g. §1¶3 says that by using adversarial training we don’t need to use task-specific losses and it does not put constraints on input modality. While true, using MSE loss is equally general). Some other claims are not supported at all or may not be true (e.g. §3.2¶1 “ResNet … aims at compressing the information in the raw observation” - to the best of my knowledge, there is no evidence for this).

To conclude, the suggested approach is not novel, the experimental evaluation is lacking, and the text contains a number of unsupported statements. I recommend to reject this paper.

[1] Lee, A.X., Zhang, R., Ebert, F., Abbeel, P., Finn, C., & Levine, S. (2018). Stochastic Adversarial Video Prediction. CoRR, abs/1804.01523.
[2] Eslami, S.M., Rezende, D.J., Besse, F., Viola, F., Morcos, A.S., Garnelo, M., Ruderman, A., Rusu, A.A., Danihelka, I., Gregor, K., Reichert, D.P., Buesing, L., Weber, T., Vinyals, O., Rosenbaum, D., Rabinowitz, N.C., King, H., Hillier, C., Botvinick, M.M., Wierstra, D., Kavukcuoglu, K., & Hassabis, D. (2018). Neural scene representation and rendering. Science, 360, 1204-1210.
[3] Denton, E.L., & Fergus, R. (2018). Stochastic Video Generation with a Learned Prior. ICML.
[4] Buesing, L., Weber, T., Racanière, S., Eslami, S.M., Rezende, D.J., Reichert, D.P., Viola, F., Besse, F., Gregor, K., Hassabis, D., & Wierstra, D. (2018). Learning and Querying Fast Generative Models for Reinforcement Learning. CoRR, abs/1802.03006.

---

### Official Review · AnonReviewer2 · 2018-11-02
**Straightforward application of existing techniques to forward modeling; experiments & writing could be improved**

**Rating:** 4
**Confidence:** 5

**Review:**

This paper proposed to train a forward model used in reinforcement learning (RL) by task-independent losses. The idea is to use the adversarial loss, infoGAN, and perception loss to replace the task-specific losses in RL.

However, the experiments did not show any benefits for the RL tasks. While it is possible that the improved prediction in terms of the Euclidean distance could lead to better results for RL, it is better to directly verify it.

Many style transfer methods can be modified to solve the problem considered in the paper. Some works on conditional GAN can also be employed. However, there is no baseline compared in the experiments.

The notations in Section 3 change from one sub-section to another. It is hard to obtain a coherent understanding about the proposed approach.

Overall, the paper identifies a key component, forward modeling, in RL and aims to improve the solution to that component. However, the proposed approach is a straightforward application of existing techniques to this problem. Both the writing and the experiments could be strengthened, per the suggestions above.

---

### Official Review · AnonReviewer3 · 2018-11-02
**Review for Deep Adversarial Forward Model**

**Rating:** 4
**Confidence:** 5

**Review:**

Summary: Model-based RL that work on pixel-based environments tend to use forward models trained with pixel-wise loss. Rather than using pixel-wise loss for an action-conditioned video prediction model ("Forward Model"), they use an adversarial loss combined with mutual-information loss (from InfoGAN) and content loss (based on difference in convnet features of VGG network, rather than pixels). They run experiments on video-action sequences collected from an Atari game (Frostbite), and on a Udacity driving dataset.

Pros: The introduction and related work section is very well written, and motivation of why one should try adversarial loss for forward models is clear.

While I think this work has potential, this paper is clearly not ready for publication, and below are a few suggestions on what I think the authors need to do to improve the work:

(1) The authors emphasize novelty, and being "first" a few times in the paper, but fail to mention the large existing work done on video prediction (i.e. [1]), many of which also used these triplet loss or adversarial losses. Sure, those works focus on video prediction, while this work focus on building a "forward model"and is supposed to be for model-based RL, but this work has not performed any model-based RL experiments, so from my point of view, it is a video-prediction model contingent on an action input. Regardless, I believe the approach and results should be compared to existing work on video prediction, and similarities and differences to existing approaches should be highlighted. Adding an action-conditioned element to existing video-prediction techniques is also fairly simple.

(2) From reading the intro/related work section, this work is clearly motivated in the direction of model-based RL, and the authors has already used this model for Frostbite. If this method is useful for model-based RL, I would expect to see experimental results for RL, at least for Frostbite (rather than just the training loss in Table 1). Rather than focusing on saying this method is the first to use triplet loss, or the first to use adversarial loss for forward models, I am much more interested in seeing a forward model that works well for RL tasks, since, that's the point right?

Although the work is promising, I can only give it a score of 4 at the moment. If the author fixes the writing to include detailed discussion with video prediction literature, with good quantitative and qualitative comparison to existing methods, that is worth 1 extra point. If the author has good results on using this forward model on environments that have previously used older forward models (such as Atari environments in [2] or CarRacing/VizDoom in [3]), and presents those results in a satisfactory way, that may increase my score by another 1-2 points depending on the depth of the experiments. Currently the paper is only < 7 pages, so I believe there is room for more substance.

Minor points:
- in related work section, should be f_{theta} not f_theta

[1] Denton et al., "Unsupervised Learning of Disentangled Representations from Video", (NIPS 2017). https://arxiv.org/abs/1705.10915
[2] https://arxiv.org/abs/1704.02254
[3] https://arxiv.org/abs/1803.10122

---

### Meta-Review · Area_Chair1 · 2018-12-14
**novelty not well justified**

**Confidence:** 5
**Recommendation:** Reject

**Metareview:**

The paper presents an action conditioned video prediction method that combines previous losses in the literature, such as, perceptual, adversarial and infogan type of losses. The reviewers point out the lack of novelty in the formulation, as well as the lack of experiments that would verify its usefulness in model based RL. There is no rebuttal thus no ground for discussion or acceptance.